# Role of Multiparametric Intestinal Ultrasound in the Evaluation of Response to Biologic Therapy in Adults with Crohn’s Disease

**DOI:** 10.3390/diagnostics12081991

**Published:** 2022-08-17

**Authors:** Pierluigi Puca, Livio Enrico Del Vecchio, Maria Elena Ainora, Antonio Gasbarrini, Franco Scaldaferri, Maria Assunta Zocco

**Affiliations:** 1IBD Unit—UOS Malattie Infiammatorie Croniche Intestinali, CEMAD, Digestive Diseases Center, Fondazione Policlinico Universitario “A. Gemelli” IRCCS, Università Cattolica del Sacro Cuore, L. Go A. Gemelli 8, 00168 Rome, Italy; 2Dipartimento di Medicina e Chirurgia Traslazionale, Università Cattolica del Sacro Cuore, L. Go F. Vito 1, 00168 Rome, Italy

**Keywords:** ultrasonography, Crohn’s disease, biologic therapy, transmural inflammation, color doppler, CEUS, elastography, SICUS, transmural healing

## Abstract

Crohn’s disease is one of the two most common types of inflammatory bowel disease. Current medical therapies are based on the use of glucocorticoids, exclusive enteral nutrition, immunosuppressors such as azathioprine and methotrexate, and biological agents such as infliximab, adalimumab, vedolizumab, or ustekinumab. International guidelines suggest regular disease assessment and surveillance through objective instruments to adjust and personalize the therapy, reducing the overall rates of hospitalization and surgery. Although endoscopy represents the gold-standard for surveillance, its frequent use is strongly bordered by associated risks and costs. Consequently, alternative non-invasive tools to objectify disease activity and rule active inflammation out are emerging. Alongside laboratory exams and computed tomography or magnetic resonance enterography, intestinal ultrasonography (IUS) shows to be a valid choice to assess transmural inflammation and to detect transmural healing, defined as bowel wall thickness normalization, no hypervascularization, normal stratification, and no creeping fat. Compared to magnetic resonance imaging (MRI) or computed tomography, CT scan, IUS is cheaper and more widespread, with very similar accuracy. Furthermore, share wave elastography, color Doppler, and contrast-enhanced ultrasonography (CEUS) succeed in amplifying the capacity to determine the disease location, disease activity, and complications. This review aimed to discuss the role of standard and novel ultrasound techniques such as CEUS, SICUS, or share wave elastography in adults with Crohn’s disease, mainly for therapeutic monitoring and follow-up.

## 1. Introduction

Crohn’s disease (CD) is a chronic relapsing condition that can affect every tract or segment of the small or large bowel, with an inflammatory, penetrating, or stricturing pattern. A peculiar feature that distinguishes CD from ulcerative colitis is the involvement of the entire bowel wall with inflammatory infiltrate throughout mucosa, submucosa, muscular layer, and within the mesentery as well [1].

Throughout the years, several molecules with different mechanisms of action have been approved for CD treatment: monoclonal antibodies acting against IL 12–23 (ustekinumab) and anti a4b7 blockers (vedolizumab)) have been added to the anti-tumor necrosis factor α (TNFα) agents [2].

Due to this wide range of therapeutic options, rising importance is being given both to the assessment of predictive factors of response to therapy and to the monitoring of response to therapies. In particular, in this field, close monitoring in a treat-to-target setting is a key principle to prevent relapse and complications Response to therapies can be classified into clinical response, defined as the improvement in symptoms and quality of life of patients; laboratory response, defined as the reduction in inflammatory markers, in particular C reactive protein (CRP) and fecal calprotectin; endoscopic response, defined as the improvement of the endoscopic markers of inflammation; transmural healing, defined as the reduction in or disappearance of wall thickening, the absence of hypervascularization, and the lack of mesenteric hypertrophy and mesenteric lymphoadenomegalies [3].

Even if the presence of clinical symptoms might reflect active inflammation, clinical scoring indices show poor correlation with the true state of disease activity. On the other hand, endoscopy, the current gold standard to objectify active inflammation, is not suitable for repeated assessment of disease activity since it is invasive, expensive, and not without risks. Hence, other objective measures are needed.

In particular, in the last few years, the assessment of transmural healing (TH) has become essential in the management of patients with CD since it has been associated with lower rates of recurrence and better outcomes [4].

Traditionally, transmural healing is evaluated by cross-sectional imaging: computed tomography (CT), magnetic resonance imaging (MRI), and intestinal ultrasound (IUS) [5]. All the mentioned methods are able to evaluate the entity and extension of wall thickening as well as the presence of strictures or dilatation of the intestinal tracts. A recent meta-analysis evidenced comparable results in the accuracy of CT, MRI, and IUS in the diagnosis of CD [6,7], even if MRI seems to be superior to IUS in recognizing CD’s relapses [8].

Among all cross-sectional imaging, IUS is a promising and non-invasive imaging technique characterized by lower costs, greater feasibility, greater comfort for patients, and lack of radiation exposure. For these reasons, in the last years, more attention has been given to the perspectives offered by this diagnostic tool. Previous studies showed high accuracy for IUS to detect disease activity and extension when compared with endoscopy or MRI with high reliability among different operators [6].

Recently, the European Crohn Colitis Organization (ECCO) released a topical review concerning the main elements to be assessed in evaluating bowel inflammation [9].

This review focused on the potential role of multiparametric IUS in predicting therapeutic efficacy of biological therapy in CD patients. The role of B-mode ultrasound, Doppler, contrast-enhanced ultrasound (CEUS), elastography, and small intestine contrast ultrasonography (SICUS) is discussed in detail, and the main studies taken into account are summarized in Table 1.

## 2. Standard Intestinal Ultrasound and Color Doppler

B-mode IUS is usually the first line imaging technique in patients with CD. It is performed using a convex probe (5–8 MHz) for panoramic view of both the small bowel and colon and subsequently a linear probe (11–14 MHz) for detailed parietal evaluation [22]. According to the current ECCO guidelines, IUS represents a fundamental tool for CD diagnosis, as well as for monitoring the disease course during follow-up [23].

The exam could provide important information about bowel wall thickness (BWT), intestinal motility, wall stiffness (through the compressibility), lumen distension (lumen diameter >2.5 cm) or strictures (lumen diameter <1 cm), signs of local inflammation (such as mesenteric fat hypertrophy or enlarged loco-regional mesenteric lymph nodes), and complications such as fistulas or abscesses [22,24].

Previous studies have shown high accuracy for IUS to detect disease activity, severity, and extent of inflammation when compared with endoscopy or MRI [25,26].

BWT is maybe the most important quantitative parameter of B-mode IUS, representing a hallmark of ultrasonography in IBD. In particular, the cut-off of 3 mm showed a sensitivity near 100% and a specificity of 83.3% in the evaluation of disease activity [27], with a good correlation with MRI and ileocolonoscopy [28,29]. Moreover, a recent study demonstrated that an increased BWT was associated with histologically assessed acute inflammation, whereas the presence of mesenteric fat hypertrophy suggested the presence of chronic inflammation [30].

Different studies evaluated the role of BWT evaluation for the follow-up of patients under biological therapy. It was demonstrated that an early reduction in BWT could be seen 2 weeks from the first dose of anti-TNF α agents [21].

BWT may play a role in predicting individual response to anti-TNF α therapy, according to a research by Saevik et al. [10]. They demonstrated that anti-TNF α monoclonal antibodies were less effective in patients with normal BWT and increased representation of the muscularis propria at baseline. Both these parameters, in fact, were considered suggestive of transmural fibrosis, a parameter associated with minor response to anti-inflammatory agents.

The occurrence of TH, considered as the normalization of all B-mode IUS parameters including a BWT < 3 mm, was proved to be linked to endoscopic mucosa healing [31,32] and predicted a better outcome in patients under anti-TNF α agents (odd ratio, OR 11.7, 95% confidence interval, CI 1.2–108.2; *p* = 0.01) [15]. Moreover, patients who reached the TH in addition to the endoscopic mucosal healing showed a lower rate and a later presentation of clinical relapse (4.5%, hazard ratio, HR 0.87, *p* = 0.01), compared to patients with only endoscopic mucosal healing [14].

In a recent trial performed in CD patients treated with ustekinumab, an improvement of B-mode IUS variables was demonstrated after only 4 weeks of therapy with a progressive normalization through week 48 [18]. A quarter of the patients enrolled (24.1%) reached a TH after 48 weeks of treatment, and B-mode IUS parameters after 4 weeks were able to predict the endoscopic response at week 48 (negative predictive value 73%)

Similarly, a multicenter study performed on 188 CD patients demonstrated an improvement of B-mode IUS parameters at 3 and 12 months after the beginning of different biological therapies (adalimumab, infliximab, vedolizumab, and ustekinumab). However, a TH was recorded only in 27.5% of patients after 12 months, with a higher rate among those treated with infliximab (37%) and with colonic involvement (40%) [33]. Moreover, an ileal BWT ≥ 4 mm was correlated with infliximab secondary non-response by a multivariate analysis in 60 CD patients (OR, 2.9; 95% CI, 1.49–5.55; *p* = 0.002) [19].

Color Doppler US adapted for slow flow detection is useful to define bowel wall flow (BWF) and to assess the semi-quantitative Limberg score [34,35]. This score estimates disease activity, with an average sensitivity of 82% [36], discriminating 4 grades of severity: grade 0 for no pathological marks, grade 1 for thickened bowel wall, grade 2 for initial vascular signals, grade 3 for bigger vascular signals, and grade 4 when these vascular signals spread over the near mesentery [22].

The role of BWF to predict the response to therapy has been evaluated essentially in the context of different activity scores also including B-mode parameters [37,38,39].

A quantitative color Doppler assessment scored by the software Color Quantification was able to predict clinical activity during anti-TNF therapy in CD patients, with a positive correlation with BWT [40].

Two recent studies using consensus panels have found a combination of BWT with BWF to accurately reflect endoscopic disease activity and endoscopic response [38,41]. In addition, a recent and partly validated scoring index with endoscopy as reference standard incorporated both BWT and BWF [42].

## 3. Small Intestine Contrast-Enhanced Ultrasonography (SICUS)

In case of stricturing or penetrating phenotypes, small intestine contrast-enhanced ultrasonography (SICUS) performed 30–40 min after the ingestion of 500–750 mL of oral contrast such as polyethylene glycol (PEG) could be very useful in patients with CD, allowing a more accurate characterization of the bowel wall and loops [43].

After the ingestion of the macrogol contrast oral solution, the contrast is observed to flow through the terminal ileum into the colon. A retrograde follow-through assessment of the entire small bowel is then performed to visualize the contrast-filled ileal and jejunal loops in a caudo-cranial sequence.

According to recent studies, SICUS improved the sensitivity of IUS for the detection of small bowel CD lesions (sensitivity 57–96% vs. 96–100%) and reduced inter-observer and intra-observer variability [44,45].

SICUS also appeared to demonstrate reasonable accuracy in detecting CD-related complications, including strictures, abscesses, and internal fistulae [46] and in the follow-up of CD patients after ileocolonic resection.

It was demonstrated that SICUS could play a role in the evaluation of response to anti-TNF α agents. In particular, an improvement or normalization of SICUS parameters, such as BWT, disease extension, phlegmon, fistulae, or abscess, predicted a better response after 1 year of therapy together with a reduction in steroid therapy and hospital admissions [16].

In another study from the same group, a new sonographic quantitative index (the sonographic lesion index for CD (SLIC)) developed to quantify changes in CD lesions detected by SICUS was used to monitor transmural bowel damage in CD patients during anti-TNF therapy. The SLIC index takes into account both the extent and severity of the small bowel damage, including stricturing and penetrating lesions as assessed by SICUS. Its features were expressed in terms of wall thickness, lesion length, lumen narrowing, and dilation and identified five classes of severity from the lower (class A) to the higher score (class E) [47]. The authors observed significant improvements in SLIC scores after induction and maintenance therapy with anti-TNFa compared with baseline values. Moreover, SLIC scores and subscores and index classes were improved significantly in patients with clinical responses [48]. Even though all SICUS studies reported and demonstrated a higher diagnostic accuracy in the detection of CD lesions compared to conventional US and a promising role in monitoring treatment disease, SICUS remains a technique with a low diffusion in the last years and a prerogative of very few centers.

## 4. Contrast-Enhanced Ultrasonography (CEUS)

Over the last few years, significant advances have occurred in improving the application of IUS in the follow-up of patients with CD. In particular, the introduction of contrast-enhanced ultrasonography (CEUS), providing a real-time visualization of the bowel microvasculature, has increased the ability to detect disease activity (Figure 1). It is based on the intravenous injections of stabilized microbubbles with gaseous content, usually sulfur hexafluoride (Sonovue, Bracco) that arrives in the intestinal wall within 10–15 s, and can be detected upon contact with US waves, which affects their size and stiffness.

In a meta-analysis including 332 CD patients, CEUS showed a pooled sensitivity of 0.94 (95% CI 0.87–0.97) and a pooled specificity of 0.79 (95% CI 0.67–0.88) in detecting active disease [49]. Similarly, CEUS was able to detect active CD in 91.1% of the cases in a study by Ripolles et al. [50].

According to several correlation studies published in the last decade, a linear association was demonstrated between CEUS and other imaging methods, such CT, MRI, or positron emission tomography (PET), traditionally used to evaluate the presence of active inflammation [51].

When CEUS is performed, a first qualitative analysis often involves the description of the pattern of enhancement. It could be outward or inward and interest part of or the whole bowel wall [52]. Moreover, a semiquantitative estimation of the ratio between the thickness of the enhanced layer and the thickness of the entire wall section may be useful. Using a range between 0.43 and 0.47 could reach a sensitivity of 81% in detecting active disease, as reported by Serra et al. [53].

The introduction of dedicated software has made it feasible to develop quantitative and objective methods of perfusion evaluation based on the construction of a time–intensity curve [54,55]. Starting from these curves it is possible to extrapolate quantitative data of bowel perfusion, such as peak intensity (PI), time to peak (Tp), mean transit time (MTT), area under the curve (AUC), and the coefficient of the enhancement wash-in slope (Pw) (Figure 2). Several studies have focused on quantitative CEUS to evaluate treatment response at different time points in patients with CD. A reduction in perfusion parameters after 12 weeks of treatment with anti-TNF alpha monoclonal antibodies has been correlated with better clinical outcomes at 1 year (85% vs. 28%; *p* = 0.0001) in 51 CD patients. Interestingly, in 9 of the 26 responders, CEUS was effective to correctly classify them based on a reduction in parietal enhancement without significant changes in BWT [11]. Quaia et al. proposed an early evaluation of time–intensity curves after only 6 weeks of pharmacological treatment and demonstrated that relative perfusion parameters derived from D-CEUS were related to treatment outcome and can separate responders from non-responders. In particular, the pretreatment values of AUC and PI and their changes at the sixth week were statistically correlated to the long-term therapeutic efficacy assessed by endoscopic evaluation within 5 months after the induction and clinical evaluation at 24 months [13]. Saevik et al. demonstrated that non-responders to steroid or anti-TNFα drugs showed a minor reduction in amplitude-based parameters (PI, AUC of wash-in, wash-in rate, and wash-out rate) after 1 month of therapy compared to responders [10]. Higher values of PI and AUC at the baseline evaluation were correlated with clinical response to biological therapy in a prospective study by Zezos et al. [56]. Based on our data, quantitative CEUS could discriminate between responders and non-responders after only two weeks of treatment with anti-TNFα [17]. In particular, a variation in four quantitative parameters extracted from time intensity curve (Pi, AUC, Pw, and MTT) after 2 weeks of treatment with infliximab was related to endoscopic response after 12 weeks. This feature could be useful to early identify patients who have more chances to benefit from treatment and to start second line therapies in the others.

Moreover, it was shown that quantitative CEUS could be helpful to identify patients with high risk of relapse after an initial benefit from therapy. In our study, patients who experienced relapse during the follow-up showed a lower decrease in PI and Pw after 2 weeks and a subsequent increase in the same parameters after 12 weeks of treatment [17].

A recent study performed in 40 patients treated with adalimumab or infliximab investigated the role of conventional IUS and CEUS parameters in predicting endoscopic response and remission early after treatment initiation [20]. Patients with endoscopic response had a significantly lower BWT after 4–8 weeks (T1) and 12–34 weeks (T2) of treatment compared to patients without endoscopic response. In particular, a decrease in BWT from baseline expressed in percentage was most accurate for prediction of endoscopic response: at T1, an 18% decrease in BWT predicted endoscopic response with 82% sensitivity and 71% specificity. Accuracy increases at T2 with a BWT decrease of 29% were most accurate to determine endoscopic response [OR: 37.50, 95% CI: 2.77–507.48, *p* = 0.006], and a cut-off value of 3.2 mm was most accurate to reflect endoscopic remission [OR: 39.42, 95% CI: 7.67–202.57, *p* < 0.0001]. Although other IUS parameters and CEUS parameters also decreased when there was endoscopic response, they were of limited merit in predicting and determining endoscopic outcomes in addition to BWT and BWF.

Beyond the classical role in the small bowel involvement, a role of CEUS in the evaluation of perianal disease through a dedicated rectal probe was also demonstrated. In a retrospective, multicenter cohort of 45 CD patients with perianal disease, CEUS demonstrated a good correlation with CT/MRI enterography to assess radiographic response after 6 and 12 weeks of ustekinumab treatment [57].

## 5. Elastography

Ultrasound elastography is a relatively new technique that is able to evaluate tissues’ stiffness and elasticity through the application of a static or dynamic stress by the ultrasonographic probe. The application of a static stress configures the so-called strain elastography (SE), whereas the application of dynamic stress is used in the shear wave elastography (SWE) [58]. A dedicated software evaluates the changes in the echo signal after compression of the target tissues and calculates how much the tissues deform according to the position of the probe. The results are shown in real-time on a color image where different degrees of strain are displayed in a color scale ranging from blue for maximum stiffness to red for maximum elasticity.

Intestinal elastography has historically been used in the diagnostic pathway of stricturing Crohn’s disease in order to distinguish inflammatory from fibrotic stenosis, a key factor in determining the correct therapeutic choice [59]. Though recent meta-analysis confirmed a good reliability in differentiating fibrotic and inflammatory strictures [60], the role of ultrasound elastography has only been evaluated in studies with small sample size and lacks a certain degree of standardization [61]. The correlation between elastosonographic parameters and histological assessment is currently under investigation. A recent systematic review showed an overall moderate to good accuracy in predicting histological fibrosis [62].

Starting from these perspectives and in consideration of the increasing number of pharmacological therapies available for CD, rising attention is being given to the possible role of elastography in monitoring the response to drugs, as well as to the research of predictive elements of response to therapies.

A pilot study performed by Chen et al. investigated the possible role of SWE in predicting early response to anti-TNFα treatment [21]. Responders presented lower baseline values of SWE compared to non-responders (15.3 kPa vs. 21.3 kPa, *p* = 0.022) and a significant reduction after 2 weeks of therapy (*p* = 0.003). Similar results were obtained by Orlando et al., where SE: patients with higher bowel stiffness (defined as a strain ratio between the mesenteric tissue and the bowel wall > or = 2) before starting anti-TNFα treatment showed higher rates of surgery during the follow-up. Moreover, baseline stiffness was lower in patients undergoing mucosal healing throughout the treatment. Finally, higher stiffness at baseline was associated with minor reduction in BWT during treatment [12].

## 6. Conclusions

Therapeutic targets in CD have changed in the last few years as the mere symptom control was not able to avoid and prevent the progressive bowel damage, but more objective parameters of inflammation are needed to improve patients’ outcomes. IUS is the object of increasing interest in monitoring of disease due to the high accuracy, large availability, low invasiveness, and relatively low cost with sensitivity and specificity comparable to CT and MRI in detecting transmural inflammation and disease complications. In particular, the application of new technologies such as quantitative CEUS and elastography improves IUS accuracy, providing a quantitative and qualitative evaluation of active inflammation and its variation during therapy. We described an approach to evaluate treatment response to biological therapy based not only on standard B-mode and Doppler US but also on CEUS to provide a quantitative, objective measurement of inflammatory activity and elastography to predict bowel stiffness.

However, the good results obtained in a research setting could be different in clinical practice, where more variables should be taken into account, such as different software, scales for the analysis of enhancement intensity, and the presence of expert and non-expert observers.

Further research and validations with different biological therapies and larger sample size are needed to validate the usefulness of IUS in clinical practice and to determine the best parameter and best timing for the assessment of response.

## Figures and Tables

**Figure 1 diagnostics-12-01991-f001:**
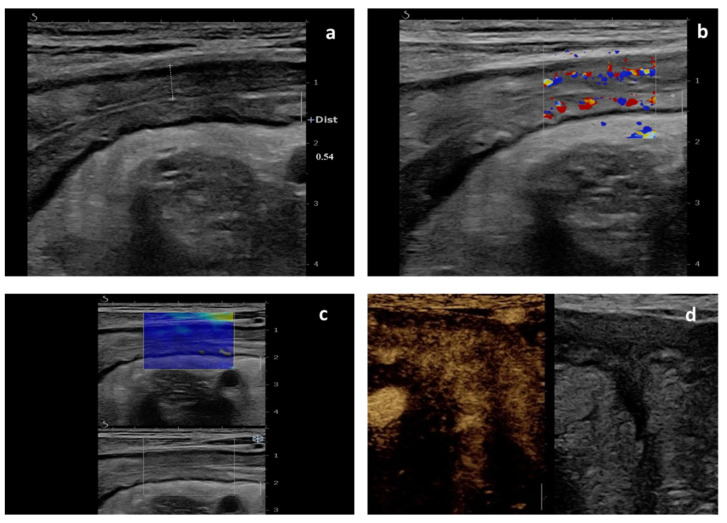
Multiparametric ultrasound evaluation in a 35-year-old man with ileal Crohn’s disease before starting biologic therapy. (**a**) B-mode ultrasound view of the terminal ileum shows thickened bowel wall (5.4 mm) with wall layer preservation. (**b**) Color Doppler images of the affected segment shows profuse mural blood flow (Limberg score 3). (**c**) Shear wave elastography of the affected segment providing a measure of tissue stiffness based on a colorimetric scale (blue is indicative of soft tissue). (**d**) Dual-screen ultrasound representation of the affected bowel, with gray-scale image (**right**) and contrast-enhanced ultrasound (CEUS) image of matching segment (**left**). CEUS image shows transmural enhancement.

**Figure 2 diagnostics-12-01991-f002:**
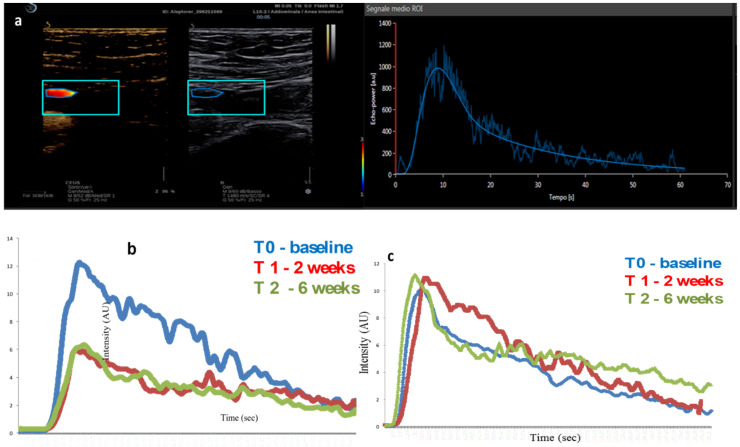
Contrast-enhanced ultrasound (CEUS) with the corresponding time–intensity curve in two patients with ileal Crohn’s disease under biological treatment. Baseline evaluation (**a**). Time–intensity curves of bowel wall enhancement at different time points showing lower enhancement and lower area under the enhancement curve after treatment in responder (**b**) and no significant difference in perfusion parameters in non-responder (**c**). Baseline = blue curve, week 2 = red curve, week 6 =green curve, AU = Arbitrary Units.

**Table 1 diagnostics-12-01991-t001:** Multiparametric ultrasound evaluation of response to biological treatment in patients with Crohn’s disease.

Article	Patients (Number)	Treatment	Ultrasound Evaluation	Results
Saevik, 2014 [10]	14	AntiTNFα	CEUS	Increased bowel perfusion after 1 month of therapy was associated with worse prognosis
Ripolles, 2016 [11]	51	AntiTNFα	BWT Color doppler flow CEUS	BWT reduction after 12 weeks of therapy but not color Doppler and CEUS parameters was associated with response to treatment
Orlando, 2018 [12]	30	AntiTNFα	SWE	Baseline lower stiffness was associated with higher probability of mucosal healing
Quaia, 2019 [13]	115	AntiTNFα	Dynamic CEUS	Pretreatment values of AUC and PI and their changes after 6 weeks of therapy were statistically correlated to long-term endoscopic and clinical efficacy
Castiglione, 2019 [14]	218	AntiTNFα	BWT	BWT < 3 after 1 year of treatment was associated to better prognosis
Paredes, 2019 [15]	36	AntiTNFα	BWT Color Doppler flow	-BWT and color Doppler flow progressively reduced during treatment-BWT < 3 after 12 weeks of treatment was associated with better outcome at 1 year
Zorzi, 2020 [16]	80	AntiTNFα	SICUS	Bwt (measured by SICUS) reduction 18 months after therapy start was associated with better long-term outcomes
Laterza, 2021 [17]	54	Infliximab	Dynamic CEUS	Variation in PI, AUC, Pw, and MTT after 2 weeks of treatment was related to endoscopic response at 12 weeks
Kucharzik, 2022 [18]	77	Ustekinumab	BWT Color Doppler flow	Absence of ultrasound response (BWT and color Doppler flow reduction) after 4 weeks of treatment predicted low endoscopic response after 1 year of treatment
Calabrese, 2022 [16]	188	Infliximab Adalimumab Vedolizumab Ustekinumab	BWT Color Doppler Flow	-Mean BWT improvement was observed from baseline up to 1 year of therapy-Greater bwt at baseline was associated with lower rates of TH at 3 months and 1 year after therapy start
Albshesh, 2022 [19]	60	Infliximab	BWT	Increased bwt (> 4 mm) during maintenance phase was associated with treatment failure
De Voogd, 2022 [20]	40	AntiTNFα	BWT Color Doppler flow	-4/8 weeks and 12/34 weeks after the beginning of therapy, BWT decrease (<3.2 mm) predicted endoscopic remission -Absence of color Doppler signal and the CEUS parameter wash-out rate improved the prediction model.
Chen, 2022 [21]	30	Infliximab	BWT, LS CEUS	-BWT and SWE at baseline were higher in non-responders -BWT, SWE, LS were reduced after 2 weeks of treatment -SWE at 2 weeks was lower in responder patients

BWT, bowel wall thickness; LS, Limberg score; CEUS, contrast-enhanced ultrasound; SWE, shear wave elastography; SICUS, small intestine contrast-enhanced ultrasound; PI, peak intensity; AUC, area under the curve; Pw, slope coefficient ow wash in; MTT, mean transit time^.^

## Data Availability

Not applicable.

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
