# Peer review of "Role of Multiparametric Intestinal Ultrasound in the Evaluation of Response to Biologic Therapy in Adults with Crohn’s Disease"

_diagnostics, 2022, doi:10.3390/diagnostics12081991_

Round 1

Reviewer 1 Report

As a GI who constantly tried to obtain transmural healing, evaluated by IUS (various methods) in patients with CD, since 2010, I truly appreciate this review and I consider it absolutely needed in clinical practice. We should always aim for transmural healing. However, before publication, I would like some points to be addressed/clarified by the authors.

 A.Title: I suggest to add “adults” (with Crohn’s disease), since most studies in children are not included. There was only 1 pediatric study, by Hakim (ref 34). Or, you choose to include also pediatric studies, which would be the best. The title (age chosen) should be also reflected in the Abstract.

B. Abstract:

1.Line 15: When the authors wrote “Current medical therapies are based on”: in order to induce remission, exclusive enteral nutrition and corticosteroids are still used. Exclusive enteral nutrition is highly recommended, with solid scientific evidence. Corticosteroids may have adverse reactions, but they are still used and recommended by the current guidelines.  Please mention also these therapies, as they are valid options, according to the guidelines.

 2.Although there has been an “agreement” over the previous years (especially starting with the wonderful paper by Castiglione- 2013) in defining transmural healing as BWT less than 3 mm, other parameters are also to be taken into consideration, nowadays, like: no hypervascularization, normal stratification, no creeping fat. Please insert. (Ref: Torsten Kucharzik, Jeroen Tielbeek, Dan Carter, Stuart A Taylor, Damian Tolan, Rune Wilkens, Robert V Bryant, Christine Hoeffel, Isabelle De Kock, Christian Maaser, Giovanni Maconi, Kerri Novak, Søren R Rafaelsen, Martina Scharitzer, Antonino Spinelli, Jordi Rimola, ECCO-ESGAR Topical Review on Optimizing Reporting for Cross-Sectional Imaging in Inflammatory Bowel Disease, Journal of Crohn's and Colitis, Volume 16, Issue 4, April 2022, Pages 523–543 Published: 10 October 2021)

 3.SICUS should me mentioned in the Abstract, as well.

 C. Key words: I would suggest to include also “elastography”, “SICUS” and “transmural healing”

D. Introduction

1. Introduction: lines 51-52: Tofacitinib is not approved for CD!!! Please correct!

2. Please define again, the correct meaning of transmural healing.

3. Please insert more recent references instead of [5] and [6].

 D. Standard Intestinal Ultrasound

1. This paragraph also includes Doppler, however it is not standard IUS and it is mentioned separately in the Abstract. Please revise.

2. Line 105: please correct to “histological”.

3. Line 148: please correct “saevdisease”.

4. Reference [60] comes after [33]. Please correct.

E. Small Intestine Contrast Enhanced Ultrasonography (SICUS)

1. Line 155: please correct to “stricturing”

2. Line 163: please delete “a” (a recent studies). Also, please correct “sensitiviry”.

 F. Contrast Enhanced Ultrasonography (CEUS)

1. Page 220: Please replace “diriment”.

 G. Elastography

Line 273: please correct to “stricturing”

Lines 282-285: “Starting from these perspectives and in consideration of the increasing number of pharmacological therapies available for CD, rising attention is being given to the possible role of elastography in monitoring the response to drugs, as well as to the research of predictive elements of response to therapies (Table 1).”. However, Table 1 contains many other studies involving IUS. They do not represent the use of elastography. Please correct and insert Table 1 in the right place in the text.

 H. Table 1: I suggest the table be made in a specific order of the studies (like the year of the study). I tried to see a logical enumeration of the studies, but I did not find it (therapy, method etc). Thank you.

 I. References: 

1. Many references are not complete, mentioning the first author only. Please correct.

2. [21] is not complete. Please insert pages 1026-1039.

3. References that should be included:

*Rimola J, Torres J, Kumar S, Taylor SA, Kucharzik T. Recent advances in clinical practice: advances in cross-sectional imaging in inflammatory bowel disease. Gut. 2022 Aug 4:gutjnl-2021-326562. doi: 10.1136/gutjnl-2021-326562

* Torsten Kucharzik, Jeroen Tielbeek, Dan Carter, Stuart A Taylor, Damian Tolan, Rune Wilkens, Robert V Bryant, Christine Hoeffel, Isabelle De Kock, Christian Maaser, Giovanni Maconi, Kerri Novak, Søren R Rafaelsen, Martina Scharitzer, Antonino Spinelli, Jordi Rimola, ECCO-ESGAR Topical Review on Optimizing Reporting for Cross-Sectional Imaging in Inflammatory Bowel Disease, Journal of Crohn's and Colitis, Volume 16, Issue 4, April 2022, Pages 523–543 Published: 10 October 2021

 J. Generally, more attention should be paid to details: there are many typos, comma is missing in many sentences, as well as the full stop. Please revise. Thank you.

Author Response

Dear reviewer, 

thank you very much for your corrections and food for thought. I have widely updated the manuscript according to your suggestions. 

In particular: 

  • I have made clear that our review is focused on adults, both in the title and in the abstract;
  • I have given a better definition of transmural healing as you suggested;
  • corrected all the typos and writing inaccuracies you highlighted; 
  • added the references you suggested; 
  • modified table 1 positioning the mentioned studies in chronological order;
  • made little modifications needed.

Speaking of references, I have used Mendeley as reference manager, using "IEEE" as citation style. 

I hope to have done a good job in implementing the manuscript with the modification needed. 

Best regards, 

Pierluigi Puca

(on the behalf of all the authors)

Reviewer 2 Report

The authors summarize the current knowledge on intestinal ultrasound in Crohn`s disease patients receiving biological therapy.  The Paper is well organized and illustrated. I have only a minor remark regarding the way of writing Crohn`s disease. Authors use different options, writing Disease or Crohn disease etc. it should be corrected; similarly ulcerative colitis is written Ulcerative Colitis. I would also recommend to supplement the table with references.

Author Response

Dear reviewer, 

thank you very much for your corrections to our work. 

As you requested, I updated the temrinology using the uniform terminology "Crohn's Disease" and "Ulcerative Colitis" throughout the entire text. 

Furthermore, I supplemented table 1 with corresponding references.

Best regards, 

Pierluigi Puca (on the behalf of all the authors)